# Development and Validation of a Low-Cost External Signal Acquisition Device for Smart Rail Pads: A Comparative Performance Study

**DOI:** 10.3390/s25061933

**Published:** 2025-03-20

**Authors:** Amparo Guillén, Fernando Moreno-Navarro, Miguel Sol-Sánchez, Guillermo R. Iglesias

**Affiliations:** 1LabIC-UGR, Laboratory of Construction Engineering, University of Granada, 18071 Granada, Spain; amguillen@ugr.es (A.G.); fmoreno@ugr.es (F.M.-N.); 2NanoMag Lab, Department of Applied Physics, Faculty of Science, University of Granada, Edificio I+D Josefina Castro, Av. de Madrid, 28, 18012 Granada, Spain; 3Biosanitary Research Institute of Granada (ibs.GRANADA), MNat Unit of Excellence, University of Granada, 18071 Granada, Spain

**Keywords:** external signal acquisition device, analog module, interactive pads, monitoring, signal acquisition, calibration

## Abstract

The development of cost-effective and reliable railway monitoring technologies is crucial for the maintenance of modern infrastructure. Embedding sensors into rail pads has emerged as a promising approach for monitoring wheel–track interactions, but the successful implementation of these systems requires a robust framework for signal data acquisition and analysis. This study validates a custom-designed External Signal Acquisition Device (ESAD) for use with smart rail pads, comparing its performance against a high-precision commercial analog module. While the commercial module delivers exceptional accuracy, its high cost, bulky size, and complex installation requirements limit its practicality for large-scale railway applications. Laboratory-scale and full-scale experiments simulating real-world railway conditions demonstrated that the custom ESAD performs comparably to the commercial module. During simulated train passages, the ESAD showed reduced signal dispersion as load and train speed increased, confirming its ability to provide reliable calibration data. Moreover, the device maintained over 95% reliability in analyzing load-to-signal linearity, ensuring consistent and dependable performance in both laboratory and field settings. However, the ESAD does have limitations, including slightly lower resolution for low frequencies and potential sensitivity to extreme environmental conditions, which may affect its performance in specific scenarios. These findings highlight the ESAD’s potential to strike a balance between cost and functionality, making it a viable solution for widespread railway monitoring applications. This research contributes to the advancement of affordable and efficient railway monitoring technologies, fostering the adoption of preventive maintenance practices and enhancing overall infrastructure performance.

## 1. Introduction

The monitoring of railway infrastructure is crucial for ensuring the safety, efficiency, and longevity of train systems. With the increasing demand of rail transport, maintaining the structural integrity of railway tracks has become more challenging. Effective monitoring systems are necessary to detect potential issues that could compromise track performance, such as wear and tear, misalignments, and stress [1,2]. Traditional methods of inspection can be costly, time-consuming, and often limited in scope [3]. Therefore, innovative solutions for continuous, real-time monitoring have gained attention in recent years, particularly those that provide accurate data at a lower cost [4,5,6].

Several researchers, as demonstrated by Jing et al. [7], have explored the design of Smart Sleepers to aid in decision-making for the maintenance and replacement of railway tracks. Similarly, Lee et al. [8] developed a wireless, cement-based sensor for self-monitoring, incorporating a wireless signal transmission module. The signal loss rate was tested based on the distance between the transmitter and receiver to assess the device’s long-term effectiveness. Additionally, fiber optic sensors are widely employed [9] in rail applications due to their adaptability to various components of the railway system such as the sensors embedded in rail pads.

Interactive pads, or Smart Pads, have emerged as promising technology for monitoring railway track conditions [10,11,12]. These devices are typically equipped with sensors that detect and transmit data related to the mechanical stresses and forces acting on the track. Several studies have demonstrated the potential of interactive pads in providing real-time information and helping track maintenance crews identify areas in need of attention before they become critical [13]. However, while these systems show great promise, the high cost of current signal acquisition devices limits their widespread adoption, especially in resource-constrained environments. This approach aligns with the work of Schalkwyk et al. (2022) [14], who introduced the concept of using “External Measuring Enclosure” devices as data acquisition systems for integration with intelligent pads. This innovation enables the creation of an integrated measurement system that minimizes costs while providing real-time and continuous monitoring of track conditions.

Robust and efficient circuit structures are essential to ensuring the accuracy and reliability of data acquisition in Smart Pads applications. Developing circuit architectures capable of processing signals under dynamic and demanding conditions is critical to advancing monitoring technologies in railway systems [15]. In this context, the speed and reliability of data acquisition are of paramount importance for ensuring track safety, optimizing maintenance, and improving operational efficiency [16,17]. Additionally, it is crucial to identify suitable methods for conducting an effective and efficient comparison between the two devices, a common practice when evaluating digital and analog systems [18]. In response to the high costs of traditional monitoring systems, recent research has focused on the development and validation of low-cost signal acquisition devices [19,20]. These devices aim to offer comparable performance to their more expensive counterparts but at a fraction of the cost. Studies have shown that low-cost alternatives can achieve similar levels of accuracy and reliability when integrated with existing monitoring technologies [21,22,23]. The advantages of these devices extend beyond cost reduction; they can also enhance the scalability of Smart Pad systems, allowing for more widespread deployment and providing more data for decision-making in track maintenance [24,25].

This study focuses on the validation and comparison of cost-effective solutions based on a low-cost signal acquisition device used in Smart Pads on railway track compared to a standard analog module device employed in laboratory-scale tests.

By evaluating its performance in train simulation tests and comparing it to an established analog module, the research aims to demonstrate the viability of affordable alternatives for real-time track monitoring. The findings suggest that, despite the lower cost, the device offers comparable accuracy and reliability, making it a suitable solution for widespread deployment in railway networks. This research addresses a critical need in railway infrastructure monitoring, particularly the development of cost-effective solutions. This aligns with the broader push for sustainable and affordable monitoring technologies in the transport sector.

## 2. Materials and Methods

### 2.1. Materials

#### 2.1.1. Smart Pads: Basic System Element

As shown in Figure 1, rail pads are positioned between the rail and the sleeper [26,27]. These pads contribute to the overall elasticity and balance of the rail infrastructure system stiffness. Their strategic surface-level contact with the rail enables direct monitoring of interactions occurring at the wheel–rail interface. Consequently, researchers are exploring the potential to integrate sensors into these pads [10,11,12,13,14]. Embedding different types of sensors (piezoelectric, piezoresistive, accelerometers, etc.) into the rail pads allows for continuous data collection on applied loads, wheel imperfections, track deflections, and other critical parameters [10,11,12,13,14].

Smart Pads are rail pads equipped with a removable sensor housing that incorporates a piezoelectric sensor. The concept involves an elastic pad designed to fit into the rail pad gap, serving as a carrier for sensors embedded within the polymeric material. This design enables the seamless integration of smart devices into the primary rail pad, with sensor types such as accelerometers, goniometers, and piezoelectric disks selected based on the specific requirements of the track. This integration allows for real-time detection of track deflections, wheel imperfections, and mechanical stresses that could compromise railway performance.

Laboratory studies and prior research [11] have demonstrated that Smart Pads equipped with piezoelectric sensors effectively detect train impact loads, track irregularities, and changes in fastening stiffness. The removable insert design enhances flexibility, allowing different types of sensors—such as accelerometers or strain gauges—to be incorporated depending on specific monitoring requirements. This modular approach ensures that Smart Pads remain adaptable to evolving railway monitoring technologies.

The core function of Smart Pads is facilitated by an embedded piezoelectric sensor, which detects dynamic forces exerted by train wheels on the track. These sensors operate based on the principle that mechanical stress, such as pressure from passing trains, deforms the piezoelectric material, generating an electrical signal proportional to the applied load. This sensor is housed within a removable polymeric insert, ensuring easy maintenance and adaptability for various rail pad designs.

The selection of quartz piezo-elements for this study is primarily due to their high stability and linearity, providing a highly linear charge response to applied force, which ensures accurate load measurements. Additionally, quartz exhibits minimal temperature sensitivity compared to other piezoelectric materials, as it has lower drift due to temperature variations. Its durability and cost-effectiveness make it a preferred choice, given that quartz elements have a long operational lifespan, making them suitable for continuous monitoring in railway environments. Furthermore, piezoelectric sensors are robust and cost-efficient compared to strain gauges or fiber-optic sensors, making them a practical choice for widespread railway applications. Another key advantage is their high sensitivity to mechanical stress, as piezoelectric materials generate an electrical charge proportional to the applied stress, making them ideal for monitoring fluctuating train loads.

To validate the performance of Smart Pads, laboratory simulations were conducted to replicate real-world railway conditions. A 250 mm UIC 54 rail section and a 357 mm prestressed concrete sleeper section were used to model a typical ballasted track structure (Figure 2a). Additionally, a 10 mm rubber mat was placed beneath the sleeper to simulate track bed elasticity. These simulations confirmed that Smart Pads accurately capture wheel–rail interaction forces, providing reliable data for track maintenance planning.

After detailing the primary components of this study, the primary objective of ESAD is illustrated in Figure 2b. Its purpose is to capture data on the loads transmitted from the train to the rail and to identify effects arising from the vehicle-rail interaction [16,28]. The sensor embedded within the pad converts these interactions into voltage signals, which are then captured in real-time by the acquisition device. This stored data can subsequently be processed and utilized to inform maintenance decisions for the track [29,30].

#### 2.1.2. Sensor Configuration and Amplification Considerations

The choice of piezoelectric material significantly impacts sensor performance, as different materials possess distinct properties that can influence the overall system behavior. Alternative materials such as Lead Zirconate Titanate (PZT), Barium Titanate (BT), and Cadmium Sulfide (CdS) each offer unique advantages and drawbacks compared to quartz as observed in Table 1. Quartz (SiO_2_) is highly stable, provides a linear response, and exhibits low temperature drift, though it has lower charge sensitivity compared to PZT. PZT, on the other hand, offers a high charge output and strong piezoelectric effect but suffers from higher temperature sensitivity, aging effects, and potential lead toxicity. BT is a lead-free option with moderate charge sensitivity, though it is less stable over time and exhibits temperature-dependent properties. CdS, commonly used in photonic applications, provides a good piezoelectric response but poses toxicity concerns and has limited mechanical durability.

If PZT were used instead of quartz, the system would exhibit higher charge sensitivity, potentially improving signal resolution for small load variations. However, temperature drift and aging effects would necessitate additional compensation techniques. BT and CdS are generally not preferred in railway applications due to their lower mechanical durability and long-term stability. For railway track monitoring, quartz remains the most reliable option due to its stability and resistance to environmental fluctuations. Nevertheless, a hybrid sensor design incorporating PZT in a thermally compensated circuit could be explored to optimize sensitivity while mitigating temperature-induced errors, albeit at an increased cost.

The ESAD system directly connects the piezoelectric sensor using a 1 MΩ resistor divider rather than an active amplifier. This design choice offers several advantages. First, it ensures lower power consumption since no active circuitry is required, making it ideal for low-power applications. Second, it introduces minimal additional noise, as active amplifiers can generate their own noise, which may degrade the signal quality. Third, it improves cost efficiency by eliminating the need for additional amplification hardware, thereby reducing overall system costs.

However, this approach also has certain limitations. One drawback is the reduced sensitivity to small signals, where weak signals may not be optimally processed, diminishing the system’s ability to capture very small load variations. Additionally, the absence of an impedance-matching circuit increases the risk of signal attenuation due to impedance mismatches. Another challenge is potential signal loss over long distances; if the sensor were positioned far from the ESAD, a pre-amplifier would be necessary to mitigate cable loss and interference.

For future implementations, integrating a low-noise charge amplifier could significantly enhance ESAD performance, particularly in scenarios requiring ultra-sensitive signal detection. However, in the current setup, the direct resistor-divider method provides a well-balanced solution in terms of simplicity, efficiency, and cost-effectiveness, making it a practical choice for railway monitoring applications.

#### 2.1.3. External Signal Acquisition Device (ESAD) and Commercial Analog Module (AM)

The External Signal Acquisition Device (ESAD) (Figure 3a) was developed as a cost-effective and reliable platform for capturing and analyzing sensor data from smart rail pads. The ESAD is powered by the Adafruit Feather M0 Adalogger (Adafruit Industries, New York, NY, USA)—an “all-in-one” board [29] designed for portable, low-power data acquisition applications. This compact and efficient microcontroller integrates essential features for railway monitoring, including USB connectivity, onboard data logging, and battery management. At its core, the Adafruit Feather M0 features the ATSAMD21 Cortex-M0 32-bit ARM processor, clocked at 48 MHz (Adafruit Industries, Brooklyn, NY, USA). This energy-efficient and versatile board provides a robust foundation for real-time data acquisition and processing [31,32].

The ESAD includes an integrated USB interface for programming, data communication, and battery charging. Real-time data are logged locally using an SD card module and transmitted to a host computer for further analysis. A 3.7 V LiPo battery powers the device, ensuring extended operational time for field applications. Additionally, an integrated charging circuit allows convenient recharging via USB. To enhance durability, all components are housed in a 3D-printed, weather-resistant enclosure that protects against environmental factors such as vibration, dust, and moisture [33,34]. The conditioned analog signals are digitized using the Feather M0’s 10-bit ADC, providing a resolution of 3 mV per step, with a sampling rate of 1 kHz to accurately capture dynamic changes during train passages.

The ESAD was tested and compared with a Commercial Analog Module (AM) (Figure 3b) under laboratory-scale conditions to evaluate its performance [35]. The AM, a high-capacity device designed for efficient analog-to-digital signal conversion, supports multiple input channels for simultaneous data acquisition. It offers a wide dynamic range for precise measurements across various signal strengths and features fast sampling rates for real-time data processing. The system operates with PCD2K software [36] (https://www.servosis.com/en/product/pcd2k/, accessed on 7 March 2025), offering a control frequency of up to 40 kHz and closed-loop control based on real analog signals. Compared to other commercial modules currently available on the market (as seen in Table 2), this module offers an advantage over the analog Metrhom module [37], as it includes a total of six auxiliary channels. The module enables simultaneous monitoring of both the sensors under test and commercial sensors, such as LVDTs, facilitating direct sensor comparison and providing more flexibility in measurement. In contrast, the Metrhom module has only two output channels, limiting its connectivity options. Another device available in the laboratory, the SIRIUS Mini [38], is a compact and portable option but comes at a higher cost and offers only four sensor connectivity channels. Additionally, the Analog Module is compatible with specialized software that integrates with various machines, allowing users to configure parameters such as data acquisition frequency, recording intervals, and data storage to tailor tests according to specific needs.

Furthermore, the alternative device was compared with the commercial analog module currently available in the construction engineering laboratory (LabIC) at the University of Granada, which is consistent with the different measuring devices installed there [39] although the validity and advantages of the ESAD are apparent, the convenience of transportation, the cost-effectiveness, the wireless connectivity, and more. Therefore, one disadvantage of the AM is its susceptibility to greater signal noise due to the 2 m cable connection. In contrast, the ESAD connects directly to the Smart Pad, minimizing noise interference. Additionally, the AM requires a wired connection, limiting flexibility, whereas the ESAD has the potential to integrate Bluetooth and Wi-Fi connectivity, enhancing functionality through edge computing for on-device data processing and reduced latency.

Figure 3c highlights two key advantages of the ESAD over the AM: its reduced cost and compact size, which enhance its adaptability to railway track systems. The economic advantage of the ESAD is significant (as observed in Figure 3c), being approximately 500 times more cost-effective than the AM. This substantial cost reduction lowers data acquisition expenses and contributes to more economical track maintenance. Furthermore, the ESAD’s size is up to 200 times smaller than the AM, enabling seamless integration within railway structures. This compact design facilitates easy transportation, handling, and installation without requiring complex placement operations.

Another notable advantage of the ESAD is its ability to connect via Bluetooth to any compatible mobile device with an appropriate application, enabling real-time measurements—an essential feature of the Smart Pads analyzed in this study. This functionality is not available in the AM, nor does it support Wi-Fi network integration as embedded in the ESAD (detailed in Figure 3a).

By prioritizing affordability, the ESAD provides an efficient solution for monitoring and maintaining track infrastructure without compromising performance. Additionally, its lower cost facilitates broader adoption and scalability across various applications, making it a practical choice for sustainable railway system management.

### 2.2. Testing Plan and Methods

Figure 4 shows that the testing plan was divided into two main stages: (i) Data processing and comparison, to compare both devices and validate the ESAD; and the (ii) Proposed calibration model to assess the accuracy and reliability of the collected data.

#### 2.2.1. Data Processing and Comparison

The first step was to validate the device through a series of comprehensive tests. These tests were designed to compare the performance of the Smart Pad under controlled conditions and simulate its behavior in realistic train operation scenarios. Both the analog module and the External Signal Acquisition Device were connected to the same Smart Pad, and identical tests were conducted for each setup to ensure consistent evaluation. This approach not only validates the functionality of the Smart Pad but also assesses the comparative performance and reliability of the two data acquisition systems.

To assess the incorporation of the ESAD, a series of tests were carried out to simulate different traffic loading conditions on the system including the Smart Pad with the signal acquisition device. As the primary function of the removable sensor insert consists of detecting variations in the wheel–rail contact, the tests used in this study stage are based on simulating different levels of load and simulating train passage. In this paper, this test is referred to as the “Frequency-Dependent Load Response Test”, adapted to the dynamic stiffness loading test according to Standard EN 13146-9 [40].

#### 2.2.2. Proposed Calibration Model

After the ESAD validation, calibration tests were conducted to ensure the accuracy of the Smart Pad when paired with both the Analog Module and the ESAD. The calibration process involved performing the Frequency-Dependent Load Response Test, which was first conducted to assess the sensor’s ability to measure and respond to varying frequencies and loads, ensuring its reliability under different operational conditions. Subsequently, the Train Load Simulation Test was performed to replicate the dynamic effects of a train passing, evaluating the sensor’s performance under simulated railway loading conditions with different frequencies and loads as well. To ensure accuracy and consistency, both tests were conducted three times using the Smart Pad equipped with the removable sensor across multiple measurement scenarios. This approach provided multiple data points for comparison. Dispersion was calculated as the ratio of the difference between the maximum and minimum values of the three measurements to their average.

To minimize potential sources of error, external factors such as temperature fluctuations, sensor misalignment, and signal noise were carefully monitored and controlled during testing. Additionally, systematic errors were addressed by ensuring consistent sensor placement and repeating measurements under identical conditions. A comparative analysis of the dispersions observed with each device (Analog Module and ESAD) across various test types further validated measurement consistency.

Introducing statistical analysis, such as standard deviation or confidence intervals, could further strengthen the evaluation of measurement reliability. Future studies may also explore real-time calibration techniques to enhance accuracy under varying operational conditions.

### 2.3. Testing Method

The Frequency-Dependent Load Response Test (Figure 5) simulating different traffic conditions was developed by adapting the tests recorded in the standard EN 13146-9 [40], which includes the assessment of the rail pads under dynamic loads. This test included 10 dynamic load steps consisting of applying 1000 cycles at 5 Hz, with stresses of 150, 300, 470, 630, 800, 950, 1200, 1600, 2000, and 2400 kPa, according to the load pressure in the rail pad area (25.200 mm^2^) which were chosen because they represent the range of loads expected for these Smart Pads during its application on tracks. In addition, to determine the susceptibility to changes in load frequency, three frequencies were studied for the 1600 kPa load (1, 5, and 10 Hz) [41].

To monitor traffic conditions, forces were applied to the rail to simulate the passage of a typical train traveling along the railway. Figure 6 illustrates the loads applied by each wheel axle and the distances between them, the application distances were determined based on each axle of the train. For example, with loads of 16.98 tons (equivalent to a train axle load), a spacing of 13.14 m is set, representing the distance between axles required to pass through the same point which depends on the train velocity. In this case, it takes a total of approximately 0.5 s (for a 13.14 m distance) for a train axle to pass a given point and for the next axle to follow, that is because the train speed was set at 100 km/h in this study to better understand the influence of variable load application from the laboratory test machine, according to the different train load levels defined in this document. However, various train axle loads and distances are observed, reflecting the different types of passing cars, from the locomotive to standard passenger cars.

This setup was designed to demonstrate the capability of the system to detect changes in contact between the train and the track. Additionally, it is important to note that the applied load represented 25% of the load exerted by a train axle (as shown in Figure 6, in the train load diagram). This percentage accounts for the load per wheel (50% of the axle load) and the concentration of stress on a central sleeper, where the load under the wheel was approximately 50% of the total—aligning with findings from [40,42].

## 3. Results and Discussion

### 3.1. Data Processing and Comparison

#### 3.1.1. Frequency-Dependent Load Response Test: Signal-Load Acquisition

Figure 7 shows a detailed demonstration of the behavior of sine waves emitted by both devices at different frequencies, the goal was to show that this representation was also accurately captured graphically by the ESAD. In this case, it represents the 15 kN load step at a 5 Hz frequency, which could be said to remain constant throughout the application of the different charges. Upon closer analysis of the waveforms for both devices, it is evident that the analog module waveform does not capture all the necessary data. This discrepancy may arise from its lower data acquisition rate for the same number of ‘peaks’. The ESAD achieves data acquisition at 10 ms, compared to 90 ms for the analog module, indicating that the ESAD is significantly more efficient in data collection. Moreover, this discrepancy between the two sine waves may be attributed to the impedance of the piezoelectric element within the Smart Rail Pad. The ESAD is directly connected to the sensor, resulting in significantly lower noise in the waves compared to the AM, which has a connection distance of approximately 2 m and relies on a cable connection that can introduce greater disturbances to the measurement wave. This highlights an advantage of the external device (ESAD), as its direct connection reduces noise in the acquired measurements.

Additionally, Figure 8 demonstrates the signal amplitude plotted at a frequency of 5 Hz across different load levels to assess each system’s capability to monitor variations in wheel–vehicle interaction. The detection of the 10 load steps, ranging from 4 kN to 60 kN, aligns precisely with the 10 detected points. The results demonstrate a linear relationship with reliability exceeding 99% in both cases, indicating that both devices could effectively detect changes in the applied rail load and, consequently, variations in fastening tension. This capability is crucial for the primary function of Smart Pads. Table 3 has been included as a reference to compare the signal values obtained from both devices. As demonstrated throughout the paper, the analysis is conducted using voltage values, which represent the signal provided by the piezoelectric sensor. Similar studies [18] comparing analog and digital modules have also utilized voltage signals from piezoelectric sensors for their comparisons. For example, under a 12 kN load, the ESAD records a voltage of 0.065 volts with a sensitivity of 2.30 kN. This indicates that the ESAD accurately measures the 12,000 newtons applied, allowing for the calculation of the load exerted on the track. For the exact same load, the AM measures 0.020 volts, slightly minor amplitude which could be due to the disturbances that occurred in the Sensor-AM connection being by a 2 m cable device compared to the direct Sensor-ESAD connection.

As shown in both Table 3 and Figure 8, the devices exhibit a greater signal jump at higher loads (40, 50, and 60 kN), as indicated by the values in Table 3 (0.208 a 0.260 the ESAD and 0.129 and 0.164 the AM in the case of the step from 40 to 50 kN) and the distance between these points in Figure 6. In other words, higher loads result in stronger signal correlation, which is advantageous for Smart Pads subjected to significant loads on the track.

#### 3.1.2. Frequency-Dependent Load Response: Various Frequencies Analysis

Figure 9 displays both Frequency-Dependent Load Test results when the Smart Pad sensor signal was processed by either the commercial Analog Module (AM) or the External Signal Acquisition Device (ESAD). At first glance, the ESAD aiming for validation provides a clear representation of each applied load step while also accommodating variations in frequency as observed in the zoom viewed in both units. Since the impact of the 5 Hz step on the sine waves was previously analyzed, Figure 9 presents individual zoomed views for both the 1 Hz and 10 Hz steps. It is evident that, at the 1 Hz step, the waves do not align as clearly as they do at the 10 Hz step, where distinct wave peaks are observed, exhibiting consistent behavior during both the loading and unloading of the sensor. Overall, the behavior of both devices is more similar at the higher frequency loading (10 Hz), since the piezoelectric sensor type (employed in this study) is more effective at higher frequencies, the data transmission would be clearer and more precise.

#### 3.1.3. Frequency-Dependent Load Response: Frequencies Analysis (Improved)

Figure 9 presents the results of the Frequency-Dependent Load Test, comparing the signal processing performance of the commercial Analog Module (AM) and the External Signal Acquisition Device (ESAD) when acquiring data from the Smart Pad sensor.

At first glance, the ESAD demonstrates a superior ability to capture and represent the applied load step, maintaining a nearly sinusoidal waveform, whereas the AM output exhibits a triangular waveform distortion. This discrepancy arises due to differences in sampling rate, signal transmission path, and noise susceptibility between the two devices.

One key advantage of the ESAD is its impact on sensor proximity and signal quality. Its direct connection to the Smart Pad significantly reduces signal degradation and noise pickup. In contrast, the AM relies on a long, unshielded 2 m cable, which increases electrical impedance and susceptibility to electromagnetic interference (EMI). This longer transmission path introduces signal distortions and attenuates higher-frequency components, leading to the observed triangular waveform in the AM output. Since the ESAD is installed closer to the sensor, it minimizes such interference and preserves the integrity of the original piezoelectric response.

Figure 9 also zooms in on the 1 Hz and 10 Hz steps to illustrate how both devices handle variations in frequency. At 1 Hz, the signal alignment is poor in both cases, as low-frequency components introduce phase lags and inconsistencies in the piezoelectric sensor’s response. Even so, the AM signal appears much clearer than ESAD. However, the importance of accurately defining the signal amplitude remains the same. This clarity makes the sine waves easier to distinguish, but the overall interpretation of the signal remains unchanged. That said, such low frequencies are rarely encountered on railroad tracks, as they are typically exposed to higher frequencies. These low-frequency components only occur at very low train speeds, which are uncommon in typical operating conditions. However, at 10 Hz, the ESAD maintains a smooth and continuous sinusoidal response, whereas the AM still struggles with distortions, highlighting its lower fidelity in capturing dynamic loads.

Overall, the ESAD consistently outperforms the AM at higher frequencies due to its superior sampling rate, reduced transmission losses, and lower impedance interference. Since Smart Pads must reliably detect rapid wheel–rail interactions, maintaining a high-quality signal across different frequencies is critical for ensuring railway monitoring accuracy.

Figure 10 illustrates the calibration comparison signals obtained from both data processing devices at the three analyzed frequencies (1, 5, and 10 Hz). Considering these three frequency steps enables the evaluation of both devices’ performance under various sensor signal behavior [23]. At first glance, the results were comparable, with both systems showing an increase in signal amplitude percentual variation as the frequency increases. When analyzing the percentage variation from 1 Hz to 5 Hz for both devices, the results were nearly identical: increasing 38% for the ESAD and 39% for the AM. However, from 5 Hz (used as a reference) to 10 Hz, the AM exhibits a more ‘linear’ behavior with an increase of 48%, while the ESAD shows a slightly lower increase of 38%. This suggests that a correlation could be established between the two devices when frequencies vary, a common scenario in train load applications, as frequency changes along with train speed.

Then, it could be concluded that one of the primary functions of the Smart Pads is detected by both devices, with the latter demonstrating greater sensitivity to higher frequencies. This finding aligns with previous studies [43], where such an increase indicates improved performance of the Smart Pads at higher frequencies—a critical requirement for railway tracks.

#### 3.1.4. Train Load Simulation Test (100 Km/h Velocity Train)

This section analyzes the behavior of the devices under varying train axel loads, applied at different load percentages. The primary objective is to compare the performance of each device and demonstrate that the ESAD reliably detects each train, as illustrated in Figure 11. In this scenario, a train traveling at 100 km/h with a full load is analyzed, focusing on the detection of each passing axle by both measuring devices. In both the ESAD and AM cases, the signal peaks transmitted by the sensor were plotted alongside the applied load. The graphs indicate that at points of higher and lower load, a corresponding reduction in signal is observed. This behavior remains consistent across both devices. For instance, under the first two applied loads (approximately 60 kN), the ESAD recorded signals of 0.5 volts, while the AM recorded signals of 0.15 volts. This difference in voltage values aligns with previous findings (as shown in Table 3), where the ESAD consistently produces higher voltage readings. This discrepancy is attributed to the application of the train wheel load occurring at a frequency of 9 Hz in this instance, where the higher the frequency of the load, the higher the signal value distinguished.

Figure 12 presents a graph comparing the ESAD and the AM, showing the measured signals for three types of trains with loads at 50%, 75%, and 100%. The results indicate very similar measurement signals between the two devices, with reliability rates of 99% for the AM and 98% for the ESAD. The number of points corresponds to the number of axles crossing (16 for each train), each with three different load levels. The “linear trend line” slopes of the two devices were also very similar, though the ESAD exhibits a slightly steeper slope (0.0118x) compared to the AM (0.003x). The slope is a critical parameter, as a steeper slope (as seen in the ESAD) enables more efficient and precise detection of changes in the stresses on the fastenings where the Smart Pads are located.

This demonstrates the reliability of both devices in accurately capturing information about the different types of passing trains, including their loads and axles, on the railway track. In terms of large-scale deployment, the ESAD offers a compact and adaptable design, making installation relatively straightforward, allowing the sensor to be attached directly to the rail pad eliminating the need to remove the pad for installation. However, potential challenges include ensuring durability in extreme weather conditions, maintaining consistent wireless communication in remote areas, and optimizing power consumption for long-term operation. Addressing these factors would be crucial for successful large-scale implementation.

### 3.2. Dispersion Analysis of Data Measurements

Figure 13 illustrates the evolution of dispersion for the same Smart Pad incorporated sensor in different measuring steps across three load steps (30, 40, and 50 kN) during a frequency cycle at 5 Hz. The analysis reveals an initial decrease in dispersion as the load increases. Notably, the ESAD exhibits lower dispersion compared to the Analog Module, underscoring its superior consistency. This reduction in dispersion, approximately 80% lower than that of the analog module, may also be influenced by the module’s sensitivity. The lower dispersion observed in the External Signal Acquisition Device (ESAD) compared to the Analog Module (AM) could be attributed to its higher sampling rate, improved signal processing, and superior hardware precision. Additionally, the external device may feature better calibration and enhanced shielding against environmental noise, all of which contribute to more stable and consistent measurements.

A small standard deviation signifies that the data points are closely clustered around the mean, indicating consistency in the measurements. In contrast, a large standard deviation shows that the data points are more widely dispersed, reflecting greater variability. This relationship helps explain the observed reduction in data dispersion when using the External Signal Acquisition Device (ESAD), as it likely enhances the consistency and reliability of the measurements [44]. Nevertheless, the analog module may be necessary in cases where the sensor requires calibration directly on the track, especially in situations where loads in critical sections are infrequent or unusual.

Additionally, Figure 14 presents an analysis of data dispersion for three different train speeds (50, 100, and 150 km/h) at three distinct load levels (50, 75 and 100%). At first glance, the highest dispersions, reaching 60%, occur at the 50% load level. As the load level increases to 100%, the dispersion decreases significantly, with values ranging between 20% and 10%. As seen in Figure 13, increasing the load results in decreased dispersion, suggesting improved measurement accuracy at higher loads. This is advantageous for the application of Smart Pads on the track, where train loads are typically substantial.

Additionally, as train speed increases, dispersion values further decrease in both cases. In other words, higher train frequencies are associated with lower dispersion, indicating enhanced performance of the sensorized device under these conditions. It is worth noting that initially, there is a significant difference in the dispersion between the ESAD and the AM. However, as the load and frequency increase, the ESAD’s values converge with those of the analog module. This could be attributed to the sensor adaptation over time, including its accommodation within the pad and the piezoelectric sensor’s ability to adjust to different sinusoidal waveforms. Additionally, the sensor improved performance at higher loads and frequencies likely contributes to this effect.

This demonstrates the robust data processing capability of the ESAD, particularly at higher loads and frequencies, as compared to the AM, which is a positive indication of its superior performance and reliability in capturing critical variations.

## 4. Conclusions

This study validates the External Signal Acquisition Device (ESAD) as a cost-effective and highly reliable alternative to traditional commercial Analog Modules (AM) for Smart Pad-based railway monitoring. Designed to address the limitations of conventional analog systems, the ESAD demonstrated superior performance in terms of signal fidelity, noise resistance, and frequency response while maintaining affordability and scalability for widespread railway infrastructure applications.

The comparative analysis between the ESAD and the AM revealed that the ESAD reliably detected all train wheel loads with an accuracy exceeding 80%, closely matching the 90% accuracy of the AM. It effectively identified individual train axles under varying load conditions and frequencies while maintaining a sinusoidal signal pattern, ensuring precise calculation of signal amplitude and confirming the sensitivity of its piezoelectric elements. Performance testing showed that the ESAD significantly reduced signal dispersion, with frequency-dependent load tests revealing up to 80% lower dispersion values compared to the AM, which exhibited dispersion rates exceeding 70%. Additionally, in train simulation experiments, the ESAD maintained its accuracy even under high-frequency and high-speed conditions, demonstrating its ability to deliver reliable, high-resolution data in real-world railway operations.

One of the key advantages of the ESAD is its direct connection to the Smart Pad sensor, which minimizes signal degradation and electrical noise. In contrast, the AM relies on a 2 m unshielded cable, making it more susceptible to signal attenuation, impedance mismatches, and electromagnetic interference (EMI). This difference in signal acquisition resulted in the ESAD producing clearer and more stable signal outputs compared to the AM, reinforcing its suitability for railway monitoring applications.

Another significant advantage of the ESAD is its integration of Bluetooth and Wi-Fi connectivity, enabling real-time data transmission and analysis through a mobile application—functionality that the AM lacks. This feature enhances the practicality of the ESAD for real-time railway monitoring, facilitating continuous data collection and improving maintenance efficiency.

Extensive laboratory testing confirmed ESAD’s accuracy, consistency, and reliability. Calibration tests involving incremental loads from 10,000 to 50,000 N established a strong correlation between the applied load and sensor output, further validating the ESAD’s ability to capture and process load variations effectively. Unlike the AM, the ESAD integrates high sensitivity, efficient data storage, and real-time processing in a compact and cost-efficient design, making it a scalable solution for continuous railway monitoring.

The findings of this study highlight the ESAD’s superiority in signal acquisition, noise reduction, and adaptability for railway track monitoring. By eliminating the need for complex wiring, reducing power consumption, and providing enhanced signal quality, the ESAD represents a breakthrough in railway monitoring technology. Its affordability, combined with its robust performance, makes it a viable alternative for large-scale railway infrastructure monitoring, ensuring better data integrity, improved predictive maintenance, and enhanced safety.

In conclusion, the ESAD stands as a powerful and practical alternative to expensive commercial analog modules, offering a more stable, accurate, and efficient solution for railway monitoring. Its ability to capture dynamic rail loads with high precision, combined with its low-cost, compact, and wireless-enabled design, positions it as a key innovation in real-time railway infrastructure assessment and maintenance.

For future research, it would be valuable to analyze the ESAD performance under extreme conditions, including variable temperatures, vibrations, and electromagnetic interference. While this study focused on design, real-scale testing on the track would provide crucial insights. Notably, the device is not suited for extreme temperatures without protective housing, as exposure could lead to damage. Additionally, exploring the integration of the sensor and the device with wireless communication, along with AI-driven data processing, could enhance efficiency and responsiveness in critical railway scenarios where long-distance measurements and rapid data analysis are essential.

## Figures and Tables

**Figure 1 sensors-25-01933-f001:**
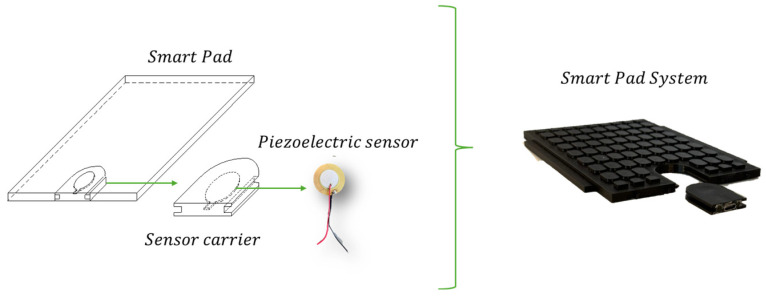
Smart Pads (Basic System Element).

**Figure 2 sensors-25-01933-f002:**
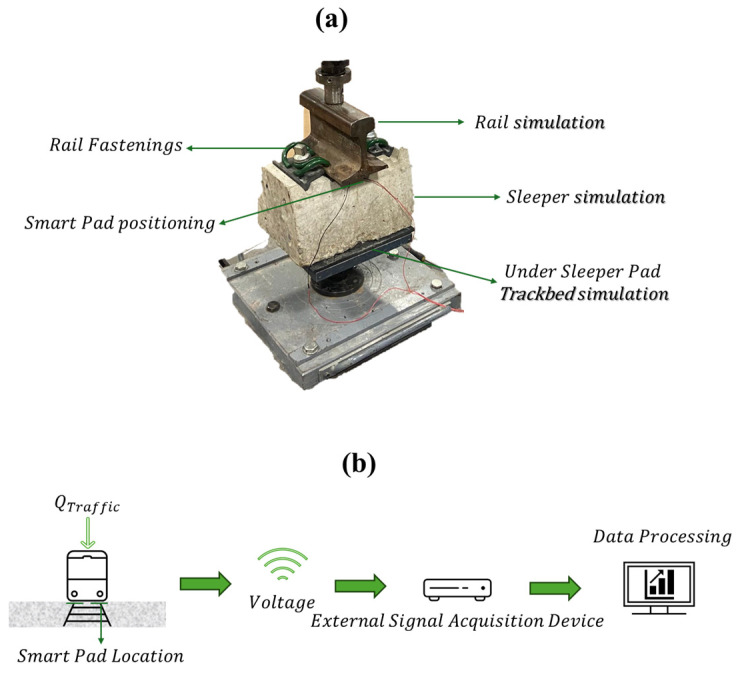
(**a**) Simulation of the composition of the railway superstructure and (**b**) Monitoring system on railway track.

**Figure 3 sensors-25-01933-f003:**
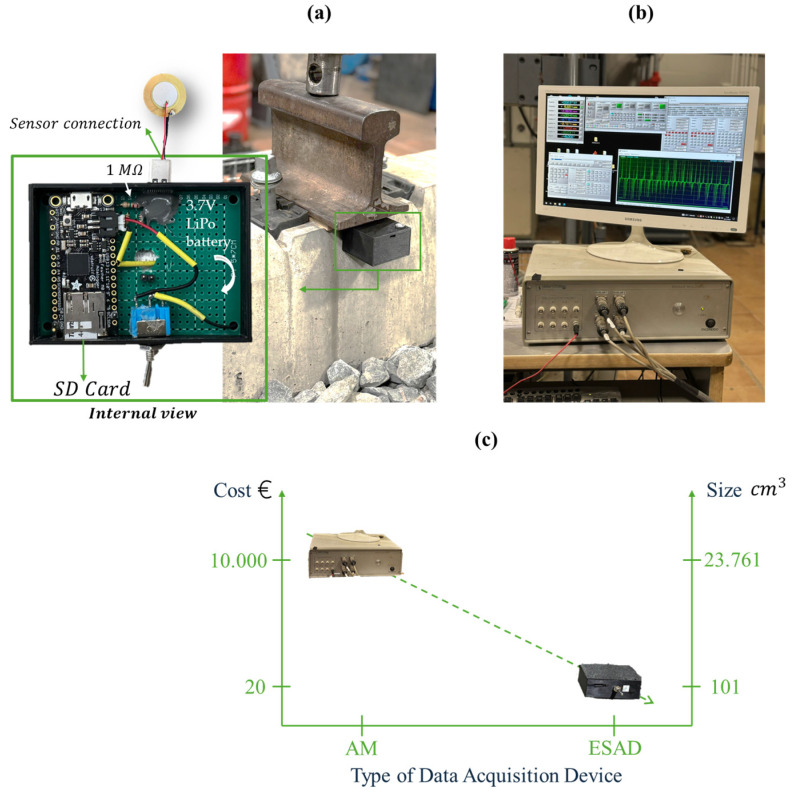
(**a**) External Signal Acquisition Device (ESAD), (**b**) Commercial Analog Module (AM) and (**c**) cost and dimension comparison between both devices.

**Figure 4 sensors-25-01933-f004:**
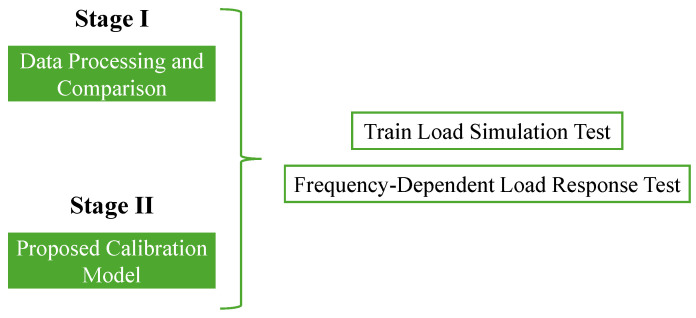
Testing Plan.

**Figure 5 sensors-25-01933-f005:**
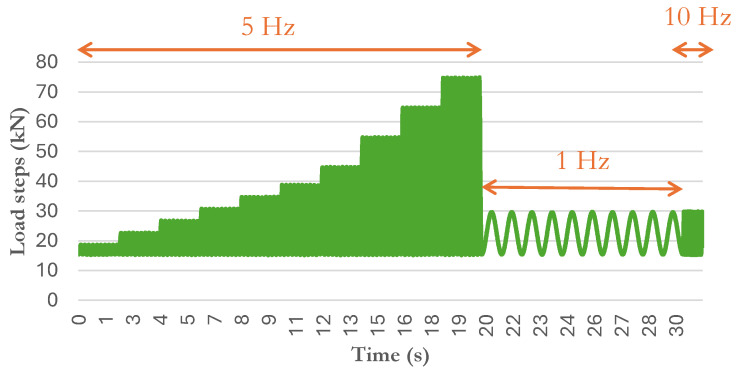
Frequency-Dependent Load Test.

**Figure 6 sensors-25-01933-f006:**
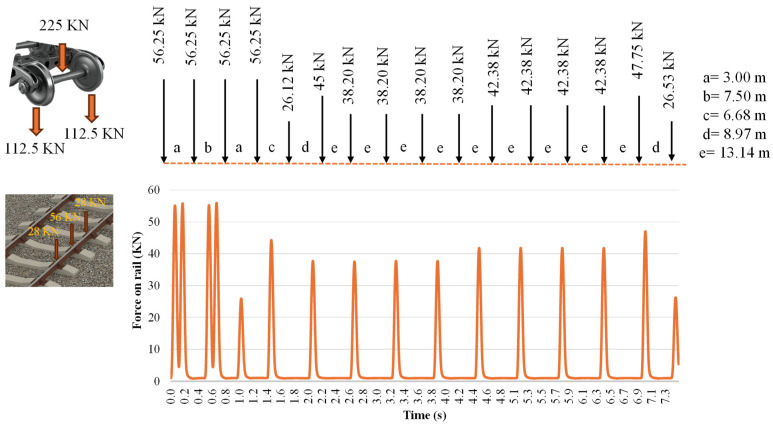
Simulation of the 100% of the train passing test. The wheel loads simulated in the laboratory (as shown in the figure above) correspond to the forces depicted in the graph below.

**Figure 7 sensors-25-01933-f007:**
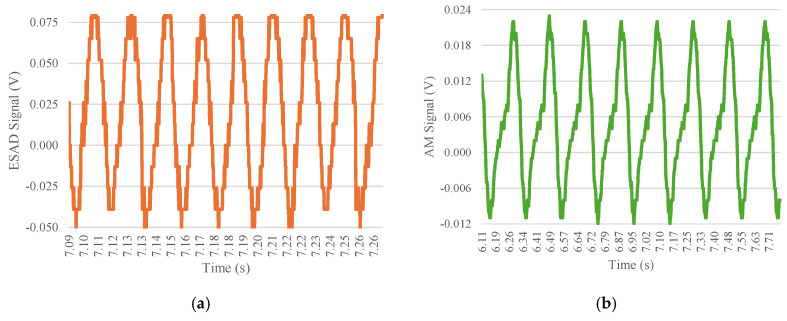
Comparative analysis of both the signal acquisition device and analog module for load step testing in 5 Hz with a 15 kN load: (**a**) External Signal Acquisition Device and (**b**) Analog Module results.

**Figure 8 sensors-25-01933-f008:**
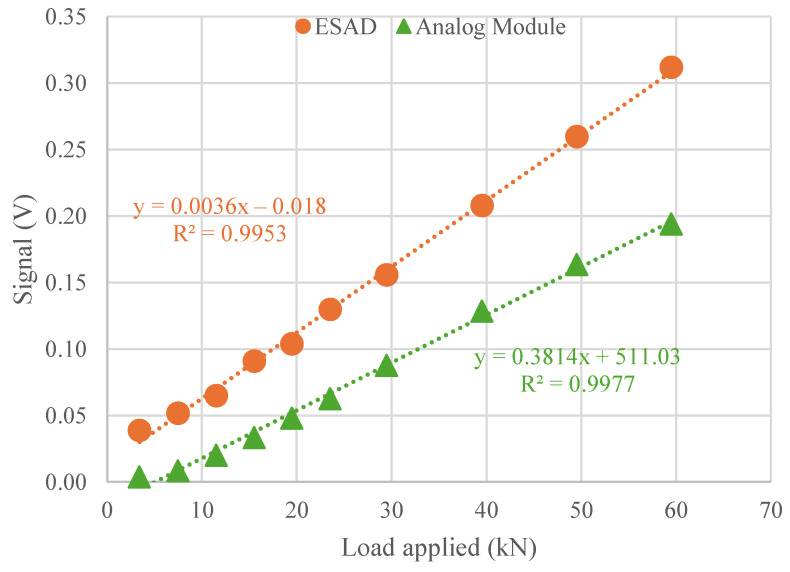
Calibration comparison for both devices on load step.

**Figure 9 sensors-25-01933-f009:**
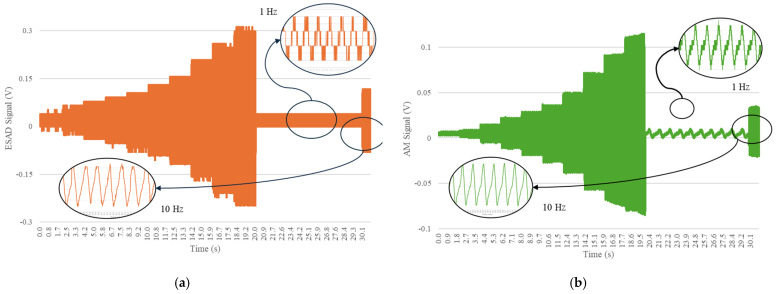
Comparative analysis of both the data acquisition device and analog module for load step testing: (**a**) External Signal Acquisition Device results and (**b**) Analog Module results.

**Figure 10 sensors-25-01933-f010:**
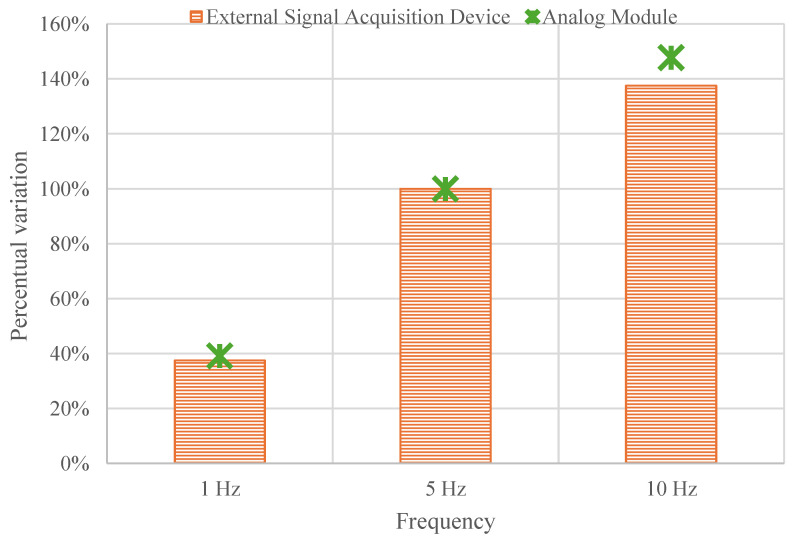
Calibration comparison for both the AM and the ESAD for three frequencies steps: 1, 5 and 10 Hz.

**Figure 11 sensors-25-01933-f011:**
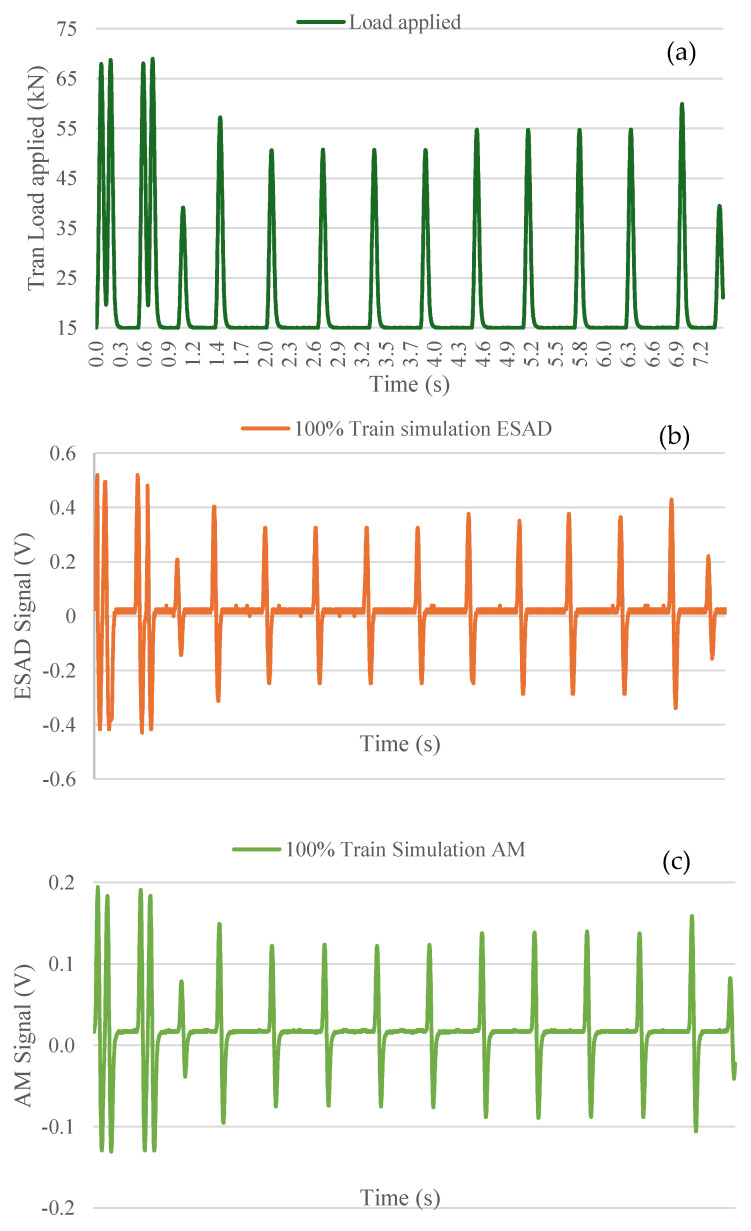
Comparative analysis of both the data acquisition device and analog module for 100% train simulation load test: (**a**) Train Load applied. (**b**) External Signal Acquisition Device and (**c**) Analog Module results.

**Figure 12 sensors-25-01933-f012:**
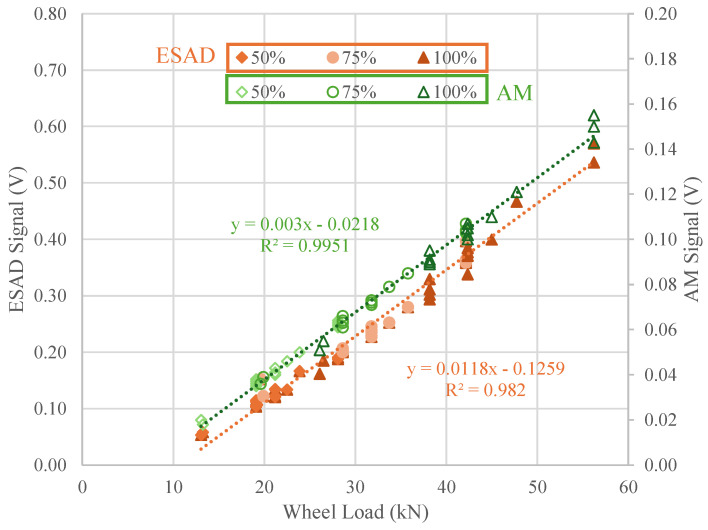
Calibration comparison for both devices on 50%, 75% and 100% train load application.

**Figure 13 sensors-25-01933-f013:**
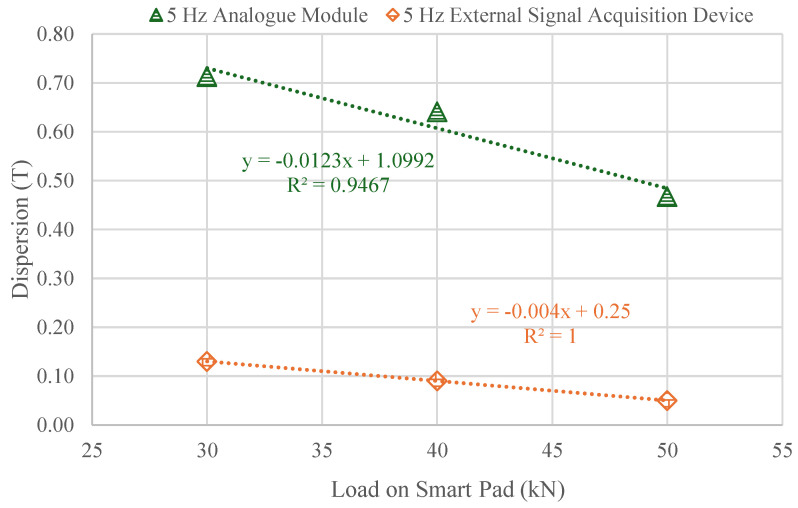
Dispersion results comparison between the Analog Module and the External Signal Acquisition Device for the Frequency-Dependent Load Response Test.

**Figure 14 sensors-25-01933-f014:**
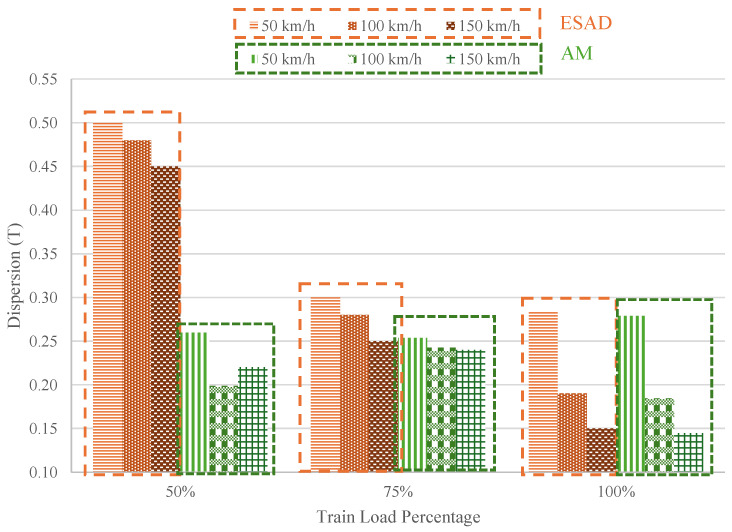
Dispersion results comparison between the Analog Module and the External Signal Acquisition Device for the Train Simulation Load Test.

**Table 1 sensors-25-01933-t001:** Summary table of material comparison.

Materials	Advantages	Disadvantages
Quartz (SiO_2_)	Highly Stable Linear Response Low Temperature drift	Lower charge sensitivity compared to PZT
Lead Zirconate Titanate (PZT)	High charge output Strong piezoelectric effect Good signal resolution	High temperature sensitivity Aging effects Potential lead toxicity
Barium Titanate (BT)	Lead-free alternative Moderate charge sensitivity	Less stable overtime Temperature-dependent properties
Cadmium Sulfide (CdS)	Good piezoelectric response Used in photonic applications	Toxicity concerns Limited mechanical durability

**Table 2 sensors-25-01933-t002:** Summary table of ESAD and AM advantages and disadvantages.

	ESAD	AM
Advantages	- Connects directly to the SmartPad - Integrated Bluetooth and Wi-fi - Low Cost - Compact size and Portable - Battery power supply - SD card storage	- Closed-loop control - Hight Cost - Control frequency of up to 40 kHz - Acquisition rate > 20 KHz (1 kHz sufficient for our needs)
Disadvantages	- SD card storage capacity - Battery autonomy - 1 kHz Acquisition rate max. - Sensitive to variations in environmental conditions. - Need insulated box	- Susceptibility to greater signal noise in the sensor and device connection - Wired connection - Need PC interface and hard disk storage - Need external supply

**Table 3 sensors-25-01933-t003:** Value comparison between both devices.

Load Applied (kN)	ESAD (V)	AM (V)
3	0.039	0.004
7	0.052	0.008
12	0.065	0.020
15	0.091	0.034
19	0.104	0.048
24	0.130	0.063
29	0.156	0.088
40	0.208	0.129
50	0.260	0.164
60	0.312	0.194

## Data Availability

The data that supports the findings of this study are available from the author upon reasonable request.

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
