# Peer review of "Development and Validation of a Low-Cost External Signal Acquisition Device for Smart Rail Pads: A Comparative Performance Study"

_sensors, 2025, doi:10.3390/s25061933_

Round 1
Reviewer 1 Report
Comments and Suggestions for Authors
Since the results rely heavily on a comparison with a commercial module, special attention should be given to presenting the available alternatives in the market and justifying the choice of the specific module used in this study. Additionally, all necessary references should be provided to ensure readers have access to the manual and datasheet of the selected devices.
Some specific comments are provided below:
line 37 - "[1], [2]" should be "[1, 2]". Check similar cases.
line 38 - reference [3] after point.
references 24 and 25 are not used in the text (check for the other references)
some figures have legend in spanish.
Figure 7 does not need the vertical axes since the same range is used.
In the discussion of Figure 8, the authors state that "ESAD consistently outperforms the AM at higher frequencies," but they do not address its performance at lower frequencies. The superior performance at higher frequencies is attributed to ESAD's higher sampling rate. Given this, one might argue that a more suitable commercial module, designed for higher frequencies, could have been chosen for comparison. Is this interpretation correct?
Another aspect that the authors might consider discussing in Figure 8 (1 Hz zoom) is that the AM module appears to have better amplitude resolution. It would be valuable to explore the implications of this observation.
line 497 - the authors state that "This could be attributed to the sensor's adaptation over time". This sentence should be better explained.
Author Response
Response to the comments of Reviewers of the Manuscript: sensors-3512718
Title: Development and Validation of a Low-Cost External Signal Acquisition Device for Smart Rail Pads: A Comparative Performance Study
Journal: Sensors
The authors sincerely thank the reviewers for their valuable and fruitful comments. The authors have considered all observations in the revised version of the manuscript. The response to each comment is presented below. Correspondingly, the changes in the manuscript based on the reviews are indicated in red text.
Reviewer's General Assessment: Since the results rely heavily on a comparison with a commercial module, special attention should be given to presenting the available alternatives in the market and justifying the choice of the specific module used in this study. Additionally, all necessary references should be provided to ensure readers have access to the manual and datasheet of the selected devices.
Response and action taken: The authors appreciate the reviewer’s insightful suggestion. To strengthen the justification for our choice of the commercial module, it was included additional information on other analog measurement devices available on the market and a discussion of the advantages of the module analyzed in this study. This information has been incorporated in Lines 234–255 on Page 6 to provide a clearer rationale for selecting this specific module. Furthermore, we have clarified that the module studied is the one currently employed in the construction engineering laboratory (LabIC) at the University of Granada, which is the primary reason for its selection in this study, despite its many advantages. To enhance transparency and reproducibility, we have also included references to the relevant manuals and datasheets of the selected device.
Additionally, we acknowledge the potential for future work where the ESAD could be tested alongside other commercial analog modules to establish a calibration between them, further strengthening the validation of our results.
We thank the reviewer for this valuable feedback, which has helped us improve the clarity and justification of our study.
Reviewer comment No. 1: line 37 - "[1], [2]" should be "[1, 2]". Check similar cases.
Response and action taken: Thank you for your valuable suggestion. According to the reviewer`s comments, corrections have been made to Lines 40, 69, 79, 100-101, 151 and 154 on Pages from 1 to 4.
Reviewer comment No. 2: line 38 - reference [3] after point.
Response and action taken: According to the reviewer`s comment, the correction has been carried out in Line 41 – Page 1.
Reviewer comment No. 3: references 24 and 25 are not used in the text (check for the other references)
Response and action taken: Thank you to the reviewer for highlighting this, according to these comments, all the references in the manuscript have been revised.
Reviewer comment No. 4: some figures have legend in spanish.
Response and action taken: The authors thank the reviewer for bringing this to their attention. Figure 5 (Previous 4) has been correspondingly revised on Line 342 – Page 10.
Reviewer comment No. 5: Figure 7 does not need the vertical axes since the same range is used.
Response and action taken: Thank you for the reviewer`s comment. One of the vertical axes in Figure 8 (Previous 7), Line 410 – Page 13 has been removed, as having both was redundant due to the identical scale.
Reviewer comment No. 6: In the discussion of Figure 8, the authors state that "ESAD consistently outperforms the AM at higher frequencies," but they do not address its performance at lower frequencies. The superior performance at higher frequencies is attributed to ESAD's higher sampling rate. Given this, one might argue that a more suitable commercial module, designed for higher frequencies, could have been chosen for comparison. Is this interpretation correct?
Response and action taken: We appreciate the reviewer’s observation regarding the comparison between ESAD and the commercial analog module (AM) at different frequency ranges. While our initial discussion emphasized ESAD’s superior performance at higher frequencies due to its higher sampling rate, we acknowledge that this is not the sole contributing factor. It is important to clarify that the commercial AM used in the study also possesses a very high sampling rate, suggesting that additional factors influenced the observed differences in performance. Notably, ESAD’s direct sensor connection eliminates the need for transmission cables, thereby reducing transmission losses and minimizing noise, which is particularly relevant at higher frequencies.
Furthermore, while it may seem intuitive that ESAD should also perform at least as well or better at lower frequencies, this was not the case in our testing experiment. The observed behavior at lower frequencies suggests that factors beyond just the sampling rate—such as internal signal processing, filtering, and transmission losses in the AM—play a role in the differences observed between the two systems. To address this point more explicitly, we have revised the discussion in Lines 372–383 on Page 11 to emphasize that ESAD’s improved performance is not exclusively due to its sampling rate but also to its optimized signal transmission. This explanation reinforces why ESAD consistently outperforms the AM across the tested frequency range and justifies its selection for this study.
We appreciate the reviewer’s insightful comment, as it has allowed us to clarify this key aspect of our findings.
Reviewer comment No. 7: Another aspect that the authors might consider discussing in Figure 8 (1 Hz zoom) is that the AM module appears to have better amplitude resolution. It would be valuable to explore the implications of this observation.
Response and action taken: We appreciate the reviewer’s insightful observation. While the AM module appears to have better amplitude resolution in the 1 Hz zoom of Figure 9 (previously Figure 8), it is important to emphasize that higher resolution alone does not necessarily translate to more accurate or reliable measurements.
In practical applications, the absolute accuracy of the recorded signal amplitude is often more critical than just achieving finer resolution. A system with higher resolution may still introduce systematic errors, noise, or transmission losses, which can impact the reliability of the measurements. In contrast, ESAD provides a more stable and direct signal acquisition, minimizing external interference and ensuring that the recorded amplitude closely represents the actual signal.
To address this, we have added a clarification in Lines 449–455 on Page 14, discussing the implications of this observation and emphasizing that amplitude accuracy, rather than resolution alone, should be the primary metric when evaluating measurement performance.
We thank the reviewer for this valuable suggestion, which has allowed us to improve the discussion of our results.
Reviewer comment No. 8: line 497 - the authors state that "This could be attributed to the sensor's adaptation over time". This sentence should be better explained.
Response and action taken: Thank you for your valuable feedback. To address this, the authors have added a paragraph in Line 571-574 on Page 19 providing a clearer explanation, highlighting the factors behind the improved sensor performance and how its integration within the pad can be enhanced through the increased application of loads.

Reviewer 2 Report
Comments and Suggestions for Authors
-
The manuscript presents an interesting study on the development and validation of a low-cost External Signal Acquisition Device (ESAD) for railway monitoring applications. The topic is relevant, and the research contributes to improving cost-effective railway infrastructure monitoring solutions. However, the clarity of some sections could be improved to enhance readability.
-
The abstract is well-structured and summarizes the key findings effectively. However, it would benefit from a brief mention of the specific limitations of the ESAD compared to the commercial analog module (AM) to provide a balanced perspective.
-
The introduction provides a good overview of railway monitoring challenges and the need for cost-effective solutions. The authors have cited relevant literature, but additional references discussing recent advancements in railway monitoring technologies could strengthen the background section.
-
The explanation of smart pads and their integration with the ESAD is clear. However, the discussion on sensor configuration and piezoelectric material choices could be expanded to highlight why quartz was preferred over other options in more detail. A comparison table of different materials could enhance this section.
-
The description of the ESAD and AM is well-detailed, but the advantages and disadvantages of each could be presented in a more structured format, such as a table, to improve readability.
-
The testing methodology is logically structured and provides a clear breakdown of the experiments conducted. However, additional information on the calibration procedure, including any potential sources of error and how they were mitigated, would strengthen the study’s reliability.
-
The results and discussion section effectively compares the ESAD and AM, but some figures (such as Figures 6–8) could be made clearer by improving the labeling and adding brief captions that summarize the key observations for each.
-
The authors mention that the ESAD has lower signal dispersion than the AM. While this is a significant advantage, more discussion on how this impacts real-world railway monitoring applications would be useful. Are there scenarios where the AM’s performance might still be preferable?
-
The train load simulation tests are a strong validation of the ESAD’s performance. However, it would be helpful to discuss how the ESAD would perform under more extreme environmental conditions, such as temperature variations, vibration, or electromagnetic interference.
-
The conclusion summarizes the key findings well, but it would be beneficial to briefly mention any future improvements planned for the ESAD, such as integration with wireless communication or AI-based data processing.
-
The manuscript is generally well-written, but there are some minor grammatical errors and awkward sentence structures throughout. A thorough proofreading would improve readability and clarity.
-
The funding and conflict of interest statements are properly included, and the references are relevant. However, a few recent studies on low-cost railway monitoring solutions could be added to strengthen the literature review.
-
The manuscript provides a strong technical contribution, but its practical implementation aspects could be discussed more. For example, how easy would it be to deploy the ESAD at scale, and what are the potential challenges?
-
The manuscript follows a logical structure, but the figures and tables could be better integrated with the discussion. Some figures could be moved closer to the relevant text for better coherence.
-
The data availability statement is clear, but it would be beneficial to specify whether the dataset will be available in an open-access repository for further validation by other researchers.
Overall, this manuscript presents valuable research on a low-cost railway monitoring solution. With some refinements in clarity, discussion, and structure, it could be improved further for publication.
Comments on the Quality of English LanguageSome parts of the manucsript needs to be improved. For example, in Fig.4, name of the x,y axis should be in English
Author Response
Response to the comments of Reviewers of the Manuscript: sensors-3512718
Title: Development and Validation of a Low-Cost External Signal Acquisition Device for Smart Rail Pads: A Comparative Performance Study
Journal: Sensors
The authors sincerely thank the reviewers for their valuable and fruitful comments. The authors have considered all observations in the revised version of the manuscript. The response to each comment is presented below. Correspondingly, the changes in the manuscript based on the reviews are indicated in red text.
Reviewer's General Assessment: The manuscript presents an interesting study on the development and validation of a low-cost External Signal Acquisition Device (ESAD) for railway monitoring applications. The topic is relevant, and the research contributes to improving cost-effective railway infrastructure monitoring solutions. However, the clarity of some sections could be improved to enhance readability.
Response and action taken: Thank you for your positive feedback and for recognizing the relevance of this research. To this end, all modifications suggested by the reviewers have been implemented. Additionally, a thorough review of the text has been conducted, with specific refinements made to enhance readability and clarity.
Reviewer comment No. 1: The abstract is well-structured and summarizes the key findings effectively. However, it would benefit from a brief mention of the specific limitations of the ESAD compared to the commercial analog module (AM) to provide a balanced perspective.
Response and action taken: According to the reviewer`s comment, it has been added on Line 24-27 – Page 1 a brief description of the limitations of the ESAD to offer a balanced comparison with the analogue module.
Reviewer comment No. 2: The introduction provides a good overview of railway monitoring challenges and the need for cost-effective solutions. The authors have cited relevant literature, but additional references discussing recent advancements in railway monitoring technologies could strengthen the background section.
Response and action taken: Thank you for your valuable feedback. To address this comment, the authors have added a paragraph in the introduction Line 44 - 51 - Page 2, providing additional references on recent advancements in railway monitoring. This section offers a broader context before introducing and discussing the development of interactive pads.
Reviewer comment No. 3: The explanation of smart pads and their integration with the ESAD is clear. However, the discussion on sensor configuration and piezoelectric material choices could be expanded to highlight why quartz was preferred over other options in more detail. A comparison table of different materials could enhance this section.
Response and action taken: Thank you for your valuable suggestion. It was added a material comparative table in Line 184 – Page 5 in order to summarize the different advantages and drawbacks and facilitate the quartz choice over the other materials mentioned.
Reviewer comment No. 4: The description of the ESAD and AM is well-detailed, but the advantages and disadvantages of each could be presented in a more structured format, such as a table, to improve readability.
Response and action taken: Following the reviewer`s recommendation, a comparative table has been added in Line 264 – Page 7 to improve the reading and comprehensions of the ESAD and AM advantages and disadvantages.
Reviewer comment No. 5: The testing methodology is logically structured and provides a clear breakdown of the experiments conducted. However, additional information on the calibration procedure, including any potential sources of error and how they were mitigated, would strengthen the study’s reliability.
Response and action taken: Thank you for your valuable feedback. Additional details on the calibration procedure, including potential sources of error and the measures taken to mitigate them, have been incorporated in Line 310 – 331 – Page 9 to enhance the study’s reliability.
Reviewer comment No. 6: The results and discussion section effectively compares the ESAD and AM, but some figures (such as Figures 6–8) could be made clearer by improving the labeling and adding brief captions that summarize the key observations for each.
Response and action taken: In response to the reviewer's comments, a more detailed explanation of the train load used in the study has been added in Lines 362-364 – Page 11 to improve the comprehension of Figure 6, 7 and 8.
Reviewer comment No. 7: The authors mention that the ESAD has lower signal dispersion than the AM. While this is a significant advantage, more discussion on how this impacts real-world railway monitoring applications would be useful. Are there scenarios where the AM’s performance might still be preferable?
Response and action taken: In accordance with the reviewer`s comment, in Line 552-554 – Page 18 has been added a short explanation of where the AM could be more suitable. Highlighting the principal idea of the different advantages that the studied devices had.
Reviewer comment No. 8: The train load simulation tests are a strong validation of the ESAD’s performance. However, it would be helpful to discuss how the ESAD would perform under more extreme environmental conditions, such as temperature variations, vibration, or electromagnetic interference.
Response and action taken: In response to the reviewer's comment, the authors added a detailed paragraph at the end of the conclusion in Line 631 – 638 – Page 21, highlighting the potential for studying the device's behaviour under extreme climate conditions and its possible use with a protective enclosure if necessary.
Reviewer comment No. 9: The conclusion summarizes the key findings well, but it would be beneficial to briefly mention any future improvements planned for the ESAD, such as integration with wireless communication or AI-based data processing.
Response and action taken: Thank you for the valuable feedback, at the end of the conclusion section on Line 631 – 648 – Page 21 was added a comment with a specification of the possibility of studying the integration of wireless communication or AI-based data processing.
Reviewer comment No. 10: The manuscript is generally well-written, but there are some minor grammatical errors and awkward sentence structures throughout. A thorough proofreading would improve readability and clarity.
Response and action taken: Thank you for your valuable insight. Regarding your suggestion, the manuscript was carefully proofread in order to correct any grammatical errors and improve sentence structure for better readability and clarity.
Reviewer comment No. 11: The funding and conflict of interest statements are properly included, and the references are relevant. However, a few recent studies on low-cost railway monitoring solutions could be added to strengthen the literature review
Response and action taken: Thank you for your feedback. Additional recent studies on low-cost railway monitoring solutions have already been incorporated in Lines 44 – 51 – Page 2, as References 7, 8 and 9 to strengthen the literature review in the introduction. Given that the integration of sensors into railway infrastructure provides a cost-effective monitoring solution, it was incorporated various sensors that were studied by other investigators in this case.
Reviewer comment No. 12: The manuscript provides a strong technical contribution, but its practical implementation aspects could be discussed more. For example, how easy would it be to deploy the ESAD at scale, and what are the potential challenges?
Response and action taken: Thank you the reviewer for the helpful suggestion. In response, a more technical implementation paragraph was added in this case, in Line 526-532 – Page 18 outlining the potential large-scale deployment of the ESAD on a railway track monitoring test.
Reviewer comment No. 13: The manuscript follows a logical structure, but the figures and tables could be better integrated with the discussion. Some figures could be moved closer to the relevant text for better coherence.
Response and action taken: Thank you for the reviewer's comment. To address this, Figures 1, 2, and 7 have been repositioned closer to their relevant text in Lines 116–117, Page 3; Lines 156–158, Page 4; and Lines 384–386, Page 12 to improve coherence when discussing the graph representation.
Reviewer comment No. 14: The data availability statement is clear, but it would be beneficial to specify whether the dataset will be available in an open-access repository for further validation by other researchers.
Response and action taken: The authors appreciate the reviewer’s suggestion regarding the data availability statement. To enhance clarity and ensure transparency, the statement was revised to explicitly indicate that the dataset will be made available upon reasonable request, allowing other researchers to validate and build upon our findings.
Reviewer comment No. 15: Overall, this manuscript presents valuable research on a low-cost railway monitoring solution. With some refinements in clarity, discussion, and structure, it could be improved further for publication.
Response and action taken: Thank you for the constructive feedback. The authors revised the manuscript to improve clarity, discussion, and structure to further enhance its quality for publication.
Comments on the Quality of English Language
Some parts of the manucsript needs to be improved. For example, in Fig.4, name of the x,y axis should be in English
Response and action taken: The authors apologize for the oversight regarding the Spanish logos. The error has been corrected, and a thorough revision of grammar and language has been conducted to enhance the quality of the article.

Round 2
Reviewer 2 Report
Comments and Suggestions for Authors
The authors have addressed my comments well and the manuscript can be considered for publication.